# Stereotactic Body Radiotherapy as a Salvage Therapy after Incomplete Radiofrequency Ablation for Hepatocellular Carcinoma: A Retrospective Propensity Score Matching Study

**DOI:** 10.3390/cancers11081116

**Published:** 2019-08-05

**Authors:** Yang-Xun Pan, Mian Xi, Yi-Zhen Fu, Dan-Dan Hu, Jun-Cheng Wang, Shi-Liang Liu, Jin-Bin Chen, Li Xu, Zhong-Guo Zhou, Meng-Zhong Liu, Min-Shan Chen, Lei Zhao, Yao-Jun Zhang

**Affiliations:** 1Sun Yat-sen University Cancer Center, State Key Laboratory of Oncology in South China, Collaborative Innovation Center for Cancer Medicine, Guangzhou 510060, China; 2Department of Liver Surgery, Sun Yat-sen University Cancer Center, Guangzhou 510060, China; 3Department of Radiation Oncology, Sun Yat-sen University Cancer Center, Guangzhou 510060, China

**Keywords:** stereotactic body radiotherapy, radiofrequency ablation, hepatocellular carcinoma, propensity score matching

## Abstract

Abstract: (1) *Background*: To investigate the clinical outcomes between radiofrequency ablation (RFA) and stereotactic body radiotherapy (SBRT) for residual hepatocellular carcinoma (RHCC). (2) *Methods*: 139 patients were diagnosed with the RHCC after post-operative checkup, among whom 39 and 33 patients underwent RFA or SBRT as salvage treatments, respectively. We applied the propensity score matching (PSM) to adjust for imbalances in treatment assignment. Local disease progression, progression-free survival (PFS), overall survival (OS), and treatment-related side effects were the study endpoints. (3) *Results*: Before PSM, the SBRT group demonstrated significantly lower local disease progression rate (6/33 vs. 23/39; *p* = 0.002), better PFS (the 1- and 3-year PFS were 63.3% and 49.3% vs. 41.5% and 22.3%, respectively, *p* = 0.036), and comparable OS (the 1- and 3-year OS were 85.4% and 71.1% vs. 97.3% and 57.6%, respectively, *p* = 0.680). After PSM of 23 matched cases, the SBRT group demonstrated significantly lower local disease progression rate, better PFS and comparable OS. Centrally located tumor predicted the worse OS. No acute grade 3+ toxicity was observed in both groups. (4) *Conclusion*: SBRT might be the preferred treatment for RHCC, especially for patients with larger tumors or tumors abutting major vessels, rather than repeated RFA.

## 1. Introduction

Hepatocellular carcinoma (HCC) is the fifth most common cancer worldwide, and the incidence of HCC has doubled in the past 20 years, making it the second global leading cause of cancer-related deaths [1,2,3]. According to guidelines of the European Society for Medical Oncology (ESMO) and American Association for the Study of Liver Disease (AASLD), liver transplantation, hepatectomy, thermal tumor ablation, and radiotherapy are the most acceptable treatment strategies for early stage HCC [4,5].

Radiofrequency ablation (RFA) induces coagulative necrosis of the tumor through thermal effect and is the first-line treatment for small HCC (≤3 cm), providing comparable long-time result with resection [6]. However, treatment with RFA poses several drawbacks that can result in incomplete ablation due to large tumor size, or with lesions located adjacent to major vessels, especially if located close to the liver hilum. Due to these limitations, the incomplete ablation rate of RFA is relatively high (varies from 2% to 60% [7,8,9,10,11]). Although repeated RFA is a common treatment of choice for residual HCC (RHCC) after RFA, the local control rate is not as satisfactory (varies from 48% to 61% [12,13]). Apart from repeated RFA, surgical resection and trans-arterial therapy are alternative treatments for RHCC patients, but these have failed to reach a significantly superior local control rate. Therefore, other potent treatment is urgently required for these patients.

Stereotactic body radiation therapy (SBRT) is a cutting-edge technology that delivers ablative radiation doses to tumors in a few fractions while minimizing the dose to normal liver tissue. Early results with SBRT have shown high local control even for large tumors or HCC which were ineligible for surgery [14]. Moreover, SBRT has been frequently used as an alternative to RFA for small HCC patients with tumors near delicate important anatomical structures or major vessels due to the heat-sink effect that can occur with RFA [15]. However, results comparing treatment efficacies between RFA and SBRT have been contradictory. Rajya guru et al. indicated that RFA yielded superior survival compared with SBRT for non-surgically managed patients with early stage HCC [16]. However, Parikh et al. reported that treatment with RFA vs. SBRT resulted in no significant difference in survival for early-stage HCC [17]. Nevertheless, Erqi et al. revealed that SBRT can be the preferred salvage therapy as the initial treatment of localized, inoperable HCC after treatment failure [18]. Since the previous results regarding on the effectiveness between SBRT and RFA were only based on the statistical model or clinical online database, real-world clinical evidence is required urgently.

In the absence of randomized data, we conducted this retrospective study to compare the effectiveness of using SBRT or RFA for treating RHCC patients who demonstrated incomplete response after an initial treatment with RFA. Furthermore, a propensity score matching (PSM) study was employed to correct for potential confounding factors that may affect the treatment assignment.

## 2. Results

### 2.1. Patient Characteristics

Between January 2008 and December 2016, 1812 patients underwent RFA, among which 139 (7.7%) patients were diagnosed as RHCC after the first RFA treatment. These patients received salvage therapies with varied modalities including liver resection, RFA, SBRT, trans-arterial chemoembolization (TACE), according to our multidisciplinary team (MDT) decision. As a result, 39 and 33 patients were treated with RFA and SBRT, respectively, with a median follow-up time of 27.2 months. Other residual cases who did not receive RFA or SBRT were excluded in our study (Figure 1).

There were no significant differences in age, gender, liver profile, Child-Pugh score, AFP level or the etiology of liver disease between the RFA and SBRT groups. The SBRT and RFA groups were significant different regarding platelet (mean, 137.0 vs. 107.0*10^9^/L; *p* = 0.038), tumor size (median, 4.1 vs. 2.1 cm in maximum diameter; *p* < 0.001), tumor location (*p* = 0.001), and follow-up period (mean, 16.30 vs. 35.30 months; *p* < 0.001). The PSM was applied and successfully matched the two treatment groups with regard to platelet counts, tumor diameter and proportion of tumor abutting major vessels. The clinical characteristics of the patients in both groups and patients that were excluded after PSM are summarized in Table 1 and Appendix A, respectively.

### 2.2. Treatment-Related Outcomes

Table 2 illustrates the post-treatment hospitalization, treatment-related side effects and treatment response in both groups. No death was seen as consequence in both groups during perioperation. Fewer adverse events were observed after RFA, and mainly constituted of pain, fever and digestive disorders. The SBRT was conducted at the outpatient clinic after pre-treatment assessment. No greater than grade 3 acute adverse effects were observed in both groups.

For RHCC patients who received repeated RFA, 1 patient suffered from incomplete response (1/39, 2.6%). This patient received a third-time RFA and had a complete response. During follow-up, 28 patients had disease progression (28/39, 71.8%), among which 23 patients suffered from the local disease progression (23/39, 59.0%). The subsequent treatments after disease progression are showed in Figure 1. One patient underwent surgical resection, 11 received a third-time RFA, 10 received TACE, 3 were prescribed with sorafenib, 1 had radiotherapy and 2 patients underwent conservative treatment. As for RHCC patients who underwent SBRT, no incomplete response was detected after the treatment and all patients were followed up to the end. During the follow-up period, 12 in 33 (12/33, 36.4%) patients had disease progression, among which 6 patients (6/33, 18.2%) had local disease progression. The subsequent treatments after disease progression are shown in Figure 1. One patient underwent surgical resection, 4 had RFA, 4 received TACE, 1 underwent radiotherapy and 2 had conservative treatment.

Collectively, compared to the RFA group, patients in the SBRT group enjoyed lower disease progression rate (28/39 vs. 12/33; *p* = 0.005), and lower local disease progression rate (23/39 vs. 6/33; *p* = 0.002). Meanwhile, the subsequent treatments after disease progression were similar (*p* = 0.073) in both groups (Table 2).

### 2.3. Progression-Free Survival before and after PSM

There was no 30-day and 90-day mortality in either group. The median PFS was 9.30 and 26.50 months for the RFA and SBRT groups, respectively. The 1-, 2- and 3-year PFS were 41.5%, 25.5% and 22.3% in the RFA group and 63.3%, 57.5% and 49.3% in the SBRT group, respectively (*p* = 0.036). Superior local control rate was observed in the SBRT group compared to the RFA group. Figure 2a shows the PFS for both groups. After PSM, for the 23 matched pairs, the median PFS was 6.20 and 26.50 months for the RFA and SBRT groups, respectively. The 1- and 2-year PFS were 34.6% and 20.7% in the RFA group and 66.4% and 56.9% in the SBRT group, respectively (Figure 2c; *p* = 0.046).

### 2.4. Overall Survival before and after PSM

21 patients in the RFA group and 6 in the SBRT group died during follow-up period. There were no significant differences in OS between the two groups. Before PSM, the 1-, 2- and 3-year OS were 97.3%, 85.4% and 57.6% in the RFA group and 85.4%, 85.4% and 71.1% in the SBRT group, respectively. The median OS estimated was 48.10 months in the RFA group, whereas the median OS was not reached in the SBRT group at the end of this study. Figure 2b demonstrates the OS for both groups. After PSM, 15 patients in the RFA group and 3 in the SBRT group died. The median OS was 33.50 months in the RFA group, whereas the median OS was not reached in the SBRT group at the end of this study. The 1-, 2- and 3-year OS were 100.0%, 88.9% and 55.6% in the RFA group and 83.7%, 83.7% and 83.7% in the SBRT group, respectively (Figure 2d).

### 2.5. Prognostic Factors

Table 3 shows the univariate and multivariate analysis for determining the prognostic factors for the PFS and OS for the entire study. Multivariate Cox regression analysis was employed by entering the univariate prognostic factors yielded from PFS and OS, including sex, tumor location, platelet, hemoglobin, and treatment type. Among these factors, centrally located tumor [hazard ratio (HR) = 2.79; 95% confidence interval (CI), 1.09–7.10; *p* = 0.032] was significantly associated to OS, indicating that tumor located in central had worse OS. Meanwhile, there were no significant prognostic factor for PFS.

## 3. Discussion

To our knowledge, this is the first study to investigate whether RHCC patients having incomplete response after an initial RFA attempt was indicated for a second RFA or referred to undergo SBRT. We found no significant difference in long-term outcomes with respect to RFA and SBRT for RHCC. However, the SBRT group demonstrated better local control rate than RFA group. Our results suggested that the initial RFA incomplete patients who had larger tumor size or tumor abutting major intrahepatic vessels would have better outcomes if offered SBRT, rather than repeated RFA.

According to the post-operative follow-up, repeated RFA provided a relatively low control rate which varied from 48% to 61% [12,19]. Another treatment strategy is urgently required in this setting. Traditionally, SBRT was regarded as a bridge therapy for orthotopic liver transplantation [20,21]. Recently, several authors have discussed the possibility of SBRT as a curative treatment in patients with HCC. Takeda et al. performed SBRT or SBRT+TACE for unresectable HCC patients [22]. They reported that SBRT possessed high local control and OS with feasible toxicities, which were confirmed by 96.3% and 66.7% for 3-year local control rate and OS rate, respectively. Daniel et al. conducted a retrospective study comparing RFA (*n* = 161) and SBRT (*n* = 63) in terms of inoperable, nonmetastatic HCC patients [14]. The SBRT had a marginally superior PFS to RFA (*p* = 0.025), and the 1- and 2-year PFS for RFA were 83.6% and 80.2% vs. 97.4% and 83.8% for SBRT, respectively. Seo et al. compared RFA with SBRT using a Markov model from online database in patients with recurrence small HCC [23]. Their obtained probability distributions of the expected OS were nearly identical with 6.452 and 6.371 years in the RFA and SBRT groups, respectively. Recently, Lee et al. reported that repeated SBRT for recurrent HCC could be safely performed with acceptable hepatic toxicity [24]. These results encouraged the use of SBRT as a comparable treatment to RFA for RHCC after RFA.

The present study compared the clinical outcomes between RFA and SBRT with respect to RHCC patients after initial RFA in terms of both PFS and OS. Patients who suffered from RHCC were able to benefit from SBRT in our study even before and after PSM. One of the possible reasons was the initial RFA incomplete response for HCC patient might imply that the lesion was unsuitable for repeated RFA treatment, excluding technical reasons. When the tumors were larger (2.10 vs. 4.10 cm; *p* < 0.001) and closer to major vessels (17/39 vs. 28/33; *p* < 0.001), the effect of RFA might be compromised because of heat loss and/or operator technique [7,9]. Moreover, the SBRT was able to deliver isodose ablative radiation to targeted area and reshape based on the tumor size, which enhanced the therapeutic effect. Additionally, the local disease progression rate was higher in RFA group also support that SBRT could provide better local control rate than that of RFA in RHCC, which was similar to other study regarding recurrent HCC [14].

Our short-term outcomes with RFA and SBRT compare favorably with the published literature before and after PSM. In the SBRT group, the largest published prospective SBRT experience for HCC (median size = 7.2 cm) from the Princess Margaret Hospital reported a 1-year local control rate of 87% for 102 patients without size dependence [25]. Smaller retrospective reports have shown that even better local control rates of more than 90% in the first year [22]. For RFA, our obtained rate of local control for RHCC were in agreement with literature reports on RFA and other initial local ablative treatments on size independence [26,27,28]. Given this concordance, we believe that the comparable short-term results with RFA and SBRT in this study are likely due to intrinsic consistency between modalities rather than unusually ineffective RFA or effective SBRT at our institution. In the present study, SBRT demonstrated favorable local control rate and better PFS. The possible explanation was that the pre-existing treatment areas might affect the design of second-time treatments, and this effect may have greater impact on RFA than SBRT, because RFA is based on a two-dimensional ultrasound as a reference of treatment. These results suggest that SBRT might be a better option for those initial RFA failure patients whose intrahepatic lesions are unfit for RFA therapy.

As for the long-term outcomes, the OS failed to demonstrate statistical significance between the two groups both before and after PSM, though SBRT demonstrated superior local control. However, the SBRT group suffered from a larger tumor volume than RFA group which indicated higher tumor burden. In contrast, RHCC in RFA group tended to have smaller tumors, better liver function scores and tumor located more peripherally which were better factors indicated better long-time outcomes in other studies [29,30]. Meanwhile, few patients reached the long-term end point in the SBRT group, which might not really reflect the long-term benefits of SBRT.

In our study, both RFA and SBRT are recommended according to the location of RHCC. For RHCC adjacent to major vessels, including portal vein and hepatic vein, SBRT enables deep and precise treatment delivery. In this setting, incomplete coagulative necrosis could occur as the thermal effect is dampened. However, SBRT is also contraindicated when the intrahepatic lesion is close to the bowel, diaphragm or heart [14,21,31]. Prospective, randomized clinical trials are needed to further investigate the best clinical scenario for both modalities; to put more emphasis on the tumor location.

Although the application of PSM analysis increased the comparability between groups, our study was retrospective, nonrandomized and relatively short follow-up time within a single center. Therefore, there may have been selection bias despite the use of PSM. Although the results were obtained by balancing tumor characteristics and liver function reserves between the two groups with PSM may improve the management of RHCC, a multicenter study should be performed to provide more solid evidence reflecting the whole picture. In addition, we lack strong evidence based on prospective randomized studies regarding the efficacy of SBRT in treating RHCC patients from RFA failure, and the role of SBRT in first-line treatment needs further study. Moreover, the follow-up time was insufficient, which could have resulted in decreasing credibility of OS and need to be expanded. Lastly, the cases volume used in this study was small and future studies with larger cohort may provide an update on the findings of the present study.

## 4. Materials and Methods

### 4.1. Patient Selection

From January 2008 to December 2016, patients who were diagnosed with RHCC, based on the presentation of classic RHCC radiologic features, at the Sun Yat-Sen University Cancer Center (SYSUCC) were enrolled in this study. The inclusion criteria were patients with: (1) initially diagnosed HCC which demonstrated incomplete response after a first round of RFA, and was detected via radiological imaging using the modified Response Evaluation Criteria in Solid Tumors (mRECST) criteria: taking the baseline of target lesion as reference, unnormal enhanced region in the arterial phase after treatment [32]; (2) liver function Child-Pugh Classification A or B; (3) tumors found to be unfeasible, difficult, or unsuitable to undergo surgery. Reasons for inoperability included central location, decreased liver function, or other comorbidities. Patients categorized with Child-Pugh Classification C, having multi-nodal disease, tumor thrombus or metastasis were excluded.

The decision to offer RFA, SBRT or other treatments for RHCC was made by liver cancer specialists through an MDT at SYSUCC, which comprised of diagnostic radiologists, pathologists, medical oncologists, radiation oncologists, surgeons, hepatologists, interventional radiologists and faculty members. Usually, surgical resection is recommended for patients with single lesion found just under liver surface. Those who suffered from RHCC with single lesion and located deeper from the liver surface were considered as candidates for RFA. Patients who had RHCCs abutted to major vessels, considered as unsuitable or refused ablation were subjected to SBRT.

This study protocol was approved by the Human Ethics Committee of SYSUCC, and informed consent was waived due to its retrospective nature (Protocol code: B2019-008-01).

### 4.2. Protocol for RFA

Contrast-enhanced ultrasonography (CEUS) was carried out for all patients before RFA. RFA was performed with the use of conscious analgesic sedation (intravenous administration of 0.1 mg of fentanyl, 5 mg of droperidol and 0.1 mg of tramadol hydrochloride) and local anesthesia (5 mL of 1% lidocaine) by an anesthesiologist. All procedures were performed percutaneously by one of three (M.-S.C.; Y.-J.Z. and L.X.) ablation experts with 6 to 15 years of experience under real-time ultrasound guidance based on previous study [33]. The ZW-II RFA system (Dalong South Technical Co., Ltd., Shenzhen, China) was used for ablation. The single-needle electrode with an exposed tip was deployed to the residual tumor bottom under ultrasound guidance while avoiding critical structures during temporary suspension of respiration. The radiofrequency generator was activated and initiated with 30 W of power. The power was increased by 10 W per minute to 60 W. Tissue impedance was continuously monitored during the ablation, and generator output was adjusted to generator maximum power or until 8 minutes had elapsed. Then, the residual lesion was rescanned to determine whether the ablative region had covered the entire tumor otherwise a second ablation was required to achieve a satisfactory ablative area. Artificial ascites or pleural effusion was used for ablating tumors on the liver surface in proximity to the diaphragm or bowel. At the end of the procedure, the needle tract was ablated to prevent bleeding and needle track seeding. After completion of the RFA procedures, the patients were sent back to the ward, their vital signs were monitored, and pain was controlled. They were discharged after close monitoring and confirmation of no bleeding or other complication and followed up regularly.

### 4.3. Protocol for SBRT

Patients underwent contrast-enhanced four-dimensional computed tomography (4DCT) scans following immobilization in the supine position. The gross tumor volume (GTV) was defined as the primary liver lesions visualized on the CT or magnetic resonance imaging (MRI), and internal target volume (ITV) was defined as the combined volume of GTVs in the 10 respiratory 4DCT phases. The planned target volume (PTV) was generated by adding a 6-mm geometric margin to the ITV. SBRT was performed by volumetric modulated arc therapy techniques, with one or two arcs using 6-MV beams. The median prescribed dose to PTV was 42 Gy (range, 30 to 54 Gy) in 6 fractions, 3 days per week. The dose constraints to 0.5 cc of the heart, esophagus, stomach, and small intestine were 40, 35, 30, and 30 Gy, respectively. The mean dose limits to normal liver and kidney were 18 and 15 Gy, respectively. The prescription dose was determined by the radiation oncologists according to the Child-Pugh Classification, tumor location, tumor size, and normal liver volume. Daily image guidance with cone-beam CT was performed before each fraction according to the standard protocol at SYSUCC. All patients underwent routine blood and liver function tests, and were evaluated clinically for SBRT-related toxicities during the course of their treatment and at periodic intervals thereafter.

### 4.4. Treatment Assessment and Follow-up

In the present study, the primary endpoint was progression-free survival (PFS) and the secondary endpoints were overall survival (OS) and adverse events. Post-procedural follow-up protocol was similar between groups and included liver profile, alpha-fetoprotein (AFP) level, and CT or MRI at the first month and every 3 months in the first year after treatment, every 4 months in the second year, and every 6 months thereafter. Adverse events were recorded according to the National Cancer Institute Common Toxicity Criteria grading system, version 5.0 during the first 30 days after treatment (acute period) [34]. PFS was defined as the absence of progressive disease by the mRECIST criteria within or at the PTV margin for patients receiving SBRT and the absence of recurrence within or adjacent to the ablation zone for patients receiving RFA [32]. Local disease progression was defined as a progressive disease detected within the treatment area. OS was defined as the interval from the time of repeated RFA or SBRT treatment to the time of death or censorship.

### 4.5. Statistical Analysis

The RFA and SBRT groups were compared at the patient level. Data on patient demographics, clinical history, laboratory findings, and RHCC tumor characteristics were collected. To avoid confounding differences in the treatment outcome due to the application of the salvage approach with those stemming from baseline differences between the RFA and SBRT groups, we estimated propensity scores by means of logistic regression and performed 1:1 nearest-neighbor individual matching based on the logit of the propensity score using a caliper width equal to 0.5 of the standard deviation without replacement [35]. The following variables were entered into the propensity model: platelet count, tumor diameter and proportion of tumor adjacent to major vessels.

With respect to baseline analysis, before PSM, *t*-tests were used for normal variables, Wilcoxon Mann-Whitney tests for ordinal but non-normal variables and *chi*-square tests for categorical variables. After PSM, paired *t*-tests and paired *chi*-square tests were used for continuous variables and categorical variables, respectively. PFS and OS were summarized with the Kaplan-Meier method. The prognostic significance of the variables in predicting PFS and OS were analyzed by univariate log-rank test and multivariate Cox proportional hazard regression models [36]. *p* < 0.05 in two-side was considered statistically significant. Analyses were performed using the R software (version 3.1.1; R Foundation for Statistical Computing, Vienna, Austria).

## 5. Conclusions

In summary, our results demonstrate that SBRT performed better local control rate and PFS, with an overall lower toxicity. Together, these findings highlight that SBRT could be the preferred alternative treatment for the initial RFA incomplete patients who had larger tumor volume or tumor abutting major intrahepatic vessels, rather than repeated RFA.

## Figures and Tables

**Figure 1 cancers-11-01116-f001:**
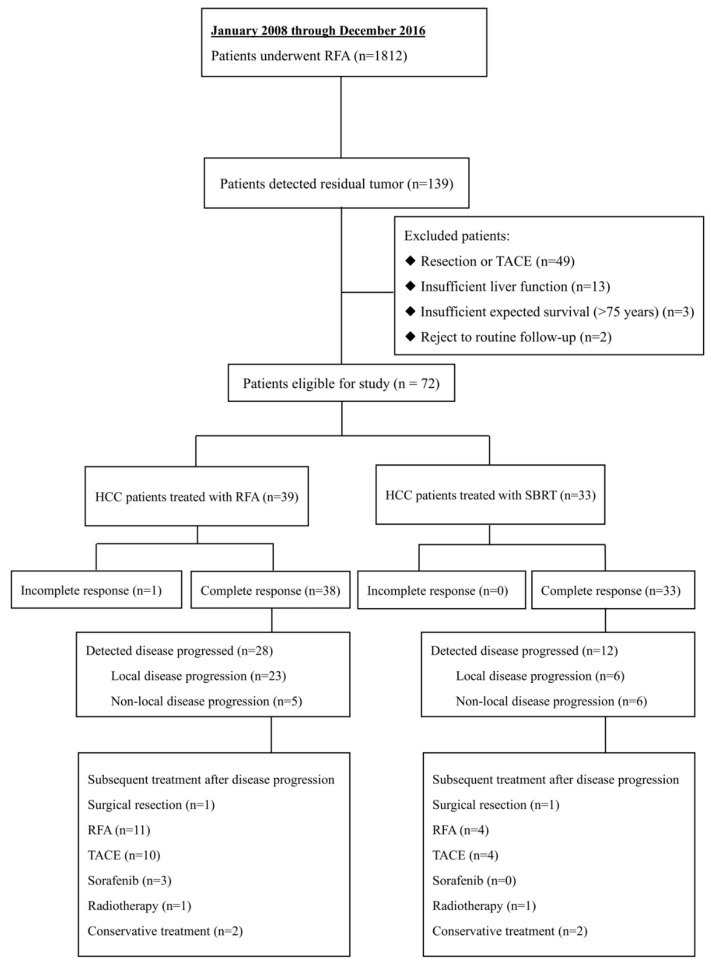
CONSORT Inclusion, exclusion criteria and treatment allocation for this retrospective study among 1812 patients with hepatocellular carcinoma (HCC) treated with radiofrequency ablation (RFA).

**Figure 2 cancers-11-01116-f002:**
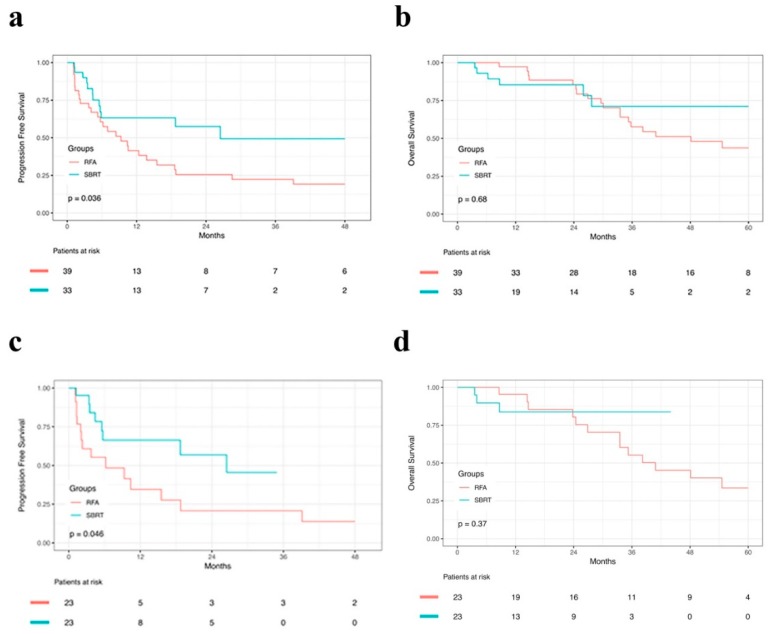
Cumulative survival curves for patients before and after propensity score matching (PSM). (**a**) Cumulative progression-free survival curves and (**b**) overall survival curves before PSM. (**c**) Cumulative progression-free survival curves and (**d**) overall survival curves after PSM.

**Table 1 cancers-11-01116-t001:** Patient and Residual Tumor Characteristics Before and After Propensity Match Score (PSM).

Variable		Before PSM			After PSM		
		RFA Group (*n* = 39)	SBRT Group (*n* = 33)	*P* Value	RFA Group (*n* = 23)	SBRT Group (*n* = 23)	*P* Value
Sex, M/F		33/6	32/1	0.137	20/3	22/1	0.601
Age, years		56.00 (49.00–64.50)	60.00 (47.00–66.00)	0.670	56.00 (48.00–65.50)	59.00 (42.50–65.50)	0.958
Cirrhosis, Y/N		22/17	15/18	0.490	11/12	10/13	1.000
B virus hepatitis, Y/N		33/6	25/8	0.517	21/2	19/4	0.662
Child-Pugh score, n	A5	37	27	0.059	22	18	0.189
	A6	1	6		1	5	
	B7	1	0		0	0	
White blood cell, *10^9^/L		5.00 (4.25–6.23)	5.19 (4.79–6.60)	0.367	5.40 (4.70–6.60)	5.19 (4.80–7.08)	0.630
Platelet, *10^9^/L		107.00 (93.00–155.00)	137.00 (113.00–192.00)	0.038	153.00 (116.00–176.00)	129.00 (100.00–188.00)	0.957
Alanine aminotransferase, U/L		33.40 (21.90–46.90)	28.50 (22.70–38.60)	0.239	33.40 (20.00–43.40)	28.50 (23.00–39.70)	0.377
Aspartate aminotransferase, U/L		32.70 (27.60–42.10)	28.40 (23.60–37.30)	0.553	32.70 (24.40–35.40)	28.40 (24.00–39.20)	0.633
Alpha fetoprotein, ng/mL		67.50 (5.34–982.00)	185.00 (4.86–840.20)	0.810	136.00 (10.00–1870.00)	218.00 (5.02–840.00)	0.878
Tumor Size, cm		2.10 (1.70–3.55)	4.10 (3.40–5.20)	<0.001	3.50 (2.35–4.00)	3.70 (3.00–4.80)	0.257
Location, peripheral/central		16/23	11/22	0.669	2/21	9/14	1.000
HCC adjacent to main vessel, Y/N		17/16	28/5	0.001	12/11	19/4	0.059
HCC abutting the capsule, Y/N		18/21	8/25	0.092	11/12	7/16	0.365
Follow–up time, months		35.30 (21.20–54.70)	16.30 (6.27–29.10)	<0.001	33.50 (19.20–50.80)	13.60 (4.13–29.10)	0.002

NOTE. Data presented as mean (interquartile range), unless otherwise noted.; Abbreviations: RFA, radiofrequency ablation; SBRT, stereotactic body radiation therapy; HCC, hepatocellular carcinoma.

**Table 2 cancers-11-01116-t002:** Treatment-Related Outcomes.

Variable		RFA Group (*n* = 39)	SBRT Group (*n* = 33)	*P* Value
Post-treatment hospitalization, days		3.00 (2.50–4.50)	-	-
Short-term treatment-related toxicities ^#^, n				
Pain		7	3	0.459
Fever		9	4	0.370
Gastrointestinal disorders		2	7	0.089
Hematological toxicity		0	2	0.401
Liver toxicities		0	2	0.401
Skin toxicities		0	1	0.933
Response according to mRECIST *, n				1.000
Complete response		38	33	
Incomplete response		1	0	
Disease progression, n		28	12	0.005
Local disease progression, n		23	6	0.002

NOTE. Data presented as mean (interquartile range), unless otherwise noted. ^#^ No greater than grade 3 toxicities effects were observed in both groups. * Tumour necrosis measurements are from 30 days post-TACE. * Tumour necrosis measurements are from 30 days post-TACE.

**Table 3 cancers-11-01116-t003:** Prognostic Factors of Progression-Free Survival and Overall Survival; Univariate Analysis (UVA) and Multivariate Analysis (MVA).

Variable		Progression-Free Survival					Overall Survival			
		Patient Number	Observed	UVA	MVA	Patient Number		Observed	UVA	MVA
Age, years	≥60	31	17	0.500	-	31		11	0.600	-
	<60	41	22			41		13		
Gender	Male	65	33	0.020	0.390	65		20	0.300	-
	Female	7	6			7		4		
Location	Peripheral	27	14	0.500	-	27		6	0.030	0.039
	Central	45	25			45		18		
Tumor size	>3 cm	42	20	0.800	-	42		13	0.400	-
	≤3 cm	30	19			30		11		
B virus hepatitis	Y	58	33	0.400	-	58		19	0.900	-
	N	14	6			14		5		
HCC adjacent to or invading main vessel	Y	45	22	0.700	-	45		11	0.200	-
	N	27	17			27		13		
HCC abutting the capsule	Y	26	15	1.000	-	26		11	0.200	-
	N	46	24			46		13		
White blood cell, *10e9/L	≥9.5	2	2	0.700	-	2		1	0.600	-
	<9.5	70	37			70		23		
Platelet, *10e9/L	≥100	51	23	0.050	0.555	51		19	0.700	-
	<100	21	16			21		5		
Hemoglobin, g/L	≥130	52	52	0.040	0.290	52		15	0.200	-
	<130	20	20			20		9		
Alanine aminotransferase, U/L	≥40	22	15	0.300	-	22		8	0.700	-
	<40	50	24			50		16		
Aspartate aminotransferase, U/L	≥45	13	7	0.800	-	13		6	0.600	-
	<45	59	32			59		18		
Albumin, g/L	≥35	68	36	0.200	-	68		23	0.800	-
	<35	4	3			4		1		
Total bilirubin, umol/dL	>34.2	4	3	0.300	-	4		2	0.200	-
	≤34.2	68	36			68		22		
Alpha fetoprotein, ng/L	≥200	31	18	0.400	-	31		9	0.900	-
	<200	41	21			41		15		
Treatment allocation	RFA	39	27	0.040	0.133	39		18	0.700	0.464
	SBRT	33	12			33		6		

Abbreviations: Y, yes; N, no; RFA, radiofrequency ablation; SBRT, stereotactic body radiation therapy.

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
