# Peer review of "Stereotactic Body Radiotherapy as a Salvage Therapy after Incomplete Radiofrequency Ablation for Hepatocellular Carcinoma: A Retrospective Propensity Score Matching Study"

_cancers, 2019, doi:10.3390/cancers11081116_

Round 1
Reviewer 1 Report
The Zhang team present a retro-perspective study on the effect of RFA and SBRT on RHCC. The authors systematically collected comprehensive diagnostic parameters for these patients enrolled in the study. Main discovery in this study is that SBRT exhibits lower local disease progression, suggesting SBRT could be the preferred alternative treatment for the initial RFA incomplete patients. I support its publication after the author consider the following concerns.
Not all data requires P value unless these two data sets are compared in parallel. Please remove these P values and just retain the ones you need to claim the differences are statistically significant.
In table 2, there must be mis-input of data or wrong calculations in the main text. For example, the author claim 29 patients had disease progression (28/39,71.8%) in line 112. But for the SBRT group the N is 33 NOT 39. I had hard time evaluate the correctness of this table, which leads to incorrect conclusion. Since this conclusion is the main discovery and novelty of this manuscript, I suggest the authors double check all original data sets in all tables. Otherwise, the manuscript will contain large amount of inconsistent results. I hope these are careless input not the flip-over of conclusions.
Author Response
Thank you for your valuable comments. Regarding the above suggestions, we made a point-to-point reply bellow.
1. We have removed the P values which are unnecessary and just retain the ones we need to claim the differences are statistically significant.
2. We apologize for our careless input of Table 2. Thanks for your kind reminder, the correct data have been double checked and illustrated both in the main text and in Table 2 (line 118-119).
Special thanks to you for your comments.
Reviewer 2 Report
The authors present a case for SBRT in patients after ablation. it is not entirely clear how there PSM approach has improved their results. They also need to specify how the method was applied to the data and which software/code was used. Do they find it concerning that OS was not
signifcantly changed? Like Feng et al in JAMA, is there any biomarker in choosing patients for
RFA or SBRT if that needs to be done again on a patient.
can the authors clarify line 71 a bit more ? "Since 70 the previous results were only based on the Markov model or clinical database, a clinical practice is required urgently."
Table column names are switched for table 2, RFA should SBRT and vice versa?
Author Response
Thank you for these valuable comments. Regarding your comments, we’d be privileged to make the following response.
1. Thank you for the comment on PSM. We have presented the PSM protocol and software in the statistical analysis section (line 338-355). In short, the R software (version 3.1.1; R Foundation for Statistical Computing, Vienna, Austria) and ‘rms’ package were used for PSM. We estimated propensity scores by means of logistic regression and performed 1:1 nearest-neighbor individual matching based on the logit of the propensity score using a caliper width equal to 0.5 of the standard deviation without replacement. And the following variables were entered into the propensity model: platelet count, tumor diameter and proportion of tumor adjacent to major vessels.
2. We appreciate the suggestion on OS, this is absolutely one of the most concerned issues regarding our clinical decisions. But it is the fact that no significance was shown before and after PSM between groups and we discussed this condition in the discussion section (line 230-238). Shortly, the SBRT group suffered from a larger tumor size than RFA group which indicated higher tumor burden. In contrast, RHCC in RFA group tended to have smaller tumors, better liver function scores and tumor located more peripherally which were better long-term outcome factors. Meanwhile, given the follow-up time was insufficient, the long-term benefits of SBRT might has been partly dampened.
3. The suggestion on biomarker is instructive, though we were not able to thoroughly investigate into the potential candidates at present, we would be most willing to include in our future large-scale studies. According to Feng et al., they believed that liver function limited the application of adaptive radiotherapy, as a result, they recommended indocyanine green retention at 15 minutes (ICGR15) as a biomarker to design the individualized adaptive radiotherapy rather than relied on population-based, tolerance to treatment. Our study discussed the appropriate treatment for RHCC based on tumor characteristics. And because of the sample limitation, we were unable to identified a biomarker to guide the treatment allocation.
4. We’ve made alterations to the unclarified wording you mentioned above, since the previous results regarding on the effectiveness between SBRT and RFA were only based on the statistical model or clinical online database, a real-world clinical evidence is required urgently (line 72-74).
5. We are sorry about the confusing part in Table 2. We have a correct illustration in the main text, but the last two rows between groups in Table 2 are reversed. A double-checked change has been made in the new version (line 118-119).
Thank you again for your valuable comments, we really appreciate the improvements we are able to make accordingly.